# Croatian National Cancer Patient Experience Survey

**DOI:** 10.3390/ijerph19148285

**Published:** 2022-07-07

**Authors:** Sandra Karabatić, Andreja Šajnić, Sanja Pleština, Marko Jakopović, Biljana Kurtović

**Affiliations:** 1Department for Respiratory Diseases Jordanovac, University Hospital Centre Zagreb, 10000 Zagreb, Croatia; asajnic@kbc-zagreb.hr (A.Š.); sanja.plestina@kbc-zagreb.hr (S.P.); marko.jakopovic@kbc-zagreb.hr (M.J.); 2Croatian Association of Patients with Lung Cancer and Other Respiratory Disease Jedra, 10000 Zagreb, Croatia; 3School of Medicine, University of Rijeka, 51000 Rijeka, Croatia; 4School of Medicine, University of Zagreb, 10000 Zagreb, Croatia; 5Department of Nursing, University of Applied Health Sciences, 10000 Zagreb, Croatia; biljana.kurtovic@zvu.hr

**Keywords:** patient cancer experience, communication, healthcare

## Abstract

Background: Cancer patients’ experiences of the healthcare system, care, and treatment are increasingly viewed as important in order to inform and improve quality of care, patient safety, and treatment efficacy. Understanding patient experience is a key step in moving toward patient-centred care. The aims of this study were to determine the experience of cancer patients in Central and Eastern European countries and to identify the needs and perspectives of oncological patients during the cancer treatment. In this paper, results from Croatia are presented. Methods: A sixty-nine item online survey was translated by native-language participating countries. Only registered members (subjects with confirmed cancer diagnosis) of the national patient oncology associations in each participating country were allowed to access and complete the online questionnaire (n = 16,458). Data were collected between October 2018 to February 2019. The Croatian Coalition of Health Associations enabled the authors of this paper to use the collected data from a sample of the Croatian participants (n = 2460) for the purposes of publication. Results: Two-thirds (67.3%) of the respondents reported satisfaction with the length of time needed for getting tests done. Bad news was delivered sensitively to 52.97% of the participants, and 52.76% received a cancer treatment plan. During the hospitalisation, 45.93% responded that they did not find someone from the hospital staff whom they could talk to about their worries and fears, and 57.48% were not given any contact information in case of concerns about their condition or treatment following the discharge. Regarding the patients’ preferences, needs, and values, 60.81% of the respondents felt that the greatest improvement would be to perform all services in one place, and 55.28% felt that improvement would be achieved through a multidisciplinary team coordinated by one person. Conclusions: The study reveals domains that need to be addressed in the overall Croatian healthcare system for oncology patients. Based on the obtained data, we can conclude that there is a large need for improvement in patient experience on the oncology pathway.

## 1. Introduction

Worldwide, an estimated 19.3 million new cancer cases, and almost 10.0 million cancer deaths occurred in 2020. Female breast cancer has surpassed lung cancer as the most diagnosed type of cancer, with an estimated 2.3 million new cases (11.7%), followed by lung (11.4%), colorectal (10%), prostate (7.3%), and stomach (5.6%) cancers. The global cancer burden is expected to amount to 28.4 million cases in 2040, a 47% rise from 2020 [1]. Patients’ experiences of the healthcare system, care, and treatment are increasingly viewed as important in order to inform and improve quality of care, patient safety, and treatment efficacy [2]. Patient experience encompasses a wide range of interactions that patients have with the healthcare system, including their care from health plans and from physicians, nurses, and staff in hospitals, physician practices, and other healthcare facilities. As an integral component of healthcare quality, patient experience includes several aspects of healthcare delivery that patients value highly when they seek and receive care, such as getting timely diagnoses, easy access to information, and good communication with healthcare providers [3]. Understanding patient experience is a key step in moving toward patient-centred care. By looking at various aspects of patient experience, one can assess the extent to which patients are receiving care that is respectful of and responsive to individual patient preferences, needs, and values. Evaluating patient experience, along with other components such as effectiveness and safety of care, is essential to providing a complete picture of healthcare quality. Throughout the world, the patient experience is recognised as an independent dimension of healthcare quality, along with clinical effectiveness and patient safety [4,5].

The Cancer Patient Experience Survey (CPES) is an annual survey which began in England (ENG) in 2010 [6]. CPES provides insight and understanding of cancer patients’ experiences of care and identifies emerging themes and areas for improvement. CPES was also conducted in Wales (WLS) in 2013 and 2016 [7,8], Northern Ireland (NIR) in 2015 and 2018 [9,10], and in Scotland (SCT) in 2016 and 2018 [11,12]. Comparison regarding a positive overall picture of cancer patients’ experience in ENG 2019 [13], NIR 2018 [10], WLS 2016 [8], and SCT 2018 [12] yielded very similar results in some domains. Additionally, surveys that were conducted in Western European countries demonstrated some variations in the domain ‘getting a timely diagnosis’: 58% in ENG, 75% in NIR, 61% in WLS, and 77% in SCT. Patients responded that they saw their general practitioner (GP) once or twice prior to being referred to a hospital on suspicion of cancer.

The aims of this study were to determine the experience of cancer patients in Central and Eastern European countries and to identify the needs and perspectives of oncological patients during the cancer treatment. In this paper, results from Croatia are presented from the Croatian National Cancer Patient Experience Survey performed in 2018/2019.

## 2. Materials and Methods

### 2.1. Respondents and Procedure

The initial online Cancer Patient Experience Survey recruited 16,458 participants from 12 countries of CEE via national patient (oncology) associations, in the period from October 2018 to February 2019. A total of 2460 Croatian cancer patients participated in the online survey. Recruitment of participants was provided by patient associations who enrolled the participants by providing links via official web pages and national patient oncology associations’ social media (Facebook). Inclusion criteria for participation in this survey were: confirmed cancer diagnosis, ability to access the internet, and the registration of those participants in their national oncology association. It was possible for caregivers to fulfil the online survey on behalf of the member of the patient association.

An interactive internet-based tool was used to gather responses illustrating the views and experiences of cancer patients. The study aimed to identify the needs and perspectives of oncological patients during the cancer treatment. In this paper, our focus was placed on Croatian cancer patients.

### 2.2. Research Instruments

The initial questionnaire, composed of 69 questions, was adapted from the England National Cancer Patient Experience Survey [6] and adjusted for Central and Eastern European countries in such a way that selected questions were grouped to reflect dimensions of patient experience and identify areas of improvement. The dimensions of patient experience and communication include experiences while receiving cancer diagnosis, perceptions, and the perceived impact of cancer on their life, including perceived support from healthcare professionals (HCPs). Selected questions included sociodemographic data (gender, age, place of residence, working status) and data on diagnosis. Questions that were selected with a focus on patient experience encompassed the following domains: getting timely appointments, easy access to information, seeking and receiving care, and communication with healthcare providers. Moreover, questions to identify suggestions and recommendations for improvement of patient care were included.

### 2.3. Data Analysis

Descriptive analysis of the data was undertaken in IBM SPSS (Version 27.0.1.0) (International Business Machine, New York, NY, USA) [14].

### 2.4. Ethical Principles

In the introductory part of the online questionnaire, the respondents were referred to the section covering the purpose of the study, and by agreeing to participate, they authorised the informed consent.

## 3. Results

Survey data were obtained from 2460 respondents from Croatia.

Table 1 provides an overview of frequency distribution according to the type of cancer; 21.91% of the respondents were male and 78.09% were female.

The youngest respondent was 4 years old and the oldest one was 83 years old, with the average age being 44.5 for the 2443 respondents who answered the age-related question. Out of 2460 respondents, almost two-thirds (60.73%) of the respondents were treated in the place of residence.

Patients’ experience in getting timely appointments.

Timely access to care is one of the six dimensions of healthcare quality [15]. Components of timely access to outpatient care include time to schedule an appointment, in-office wait time, and the timing of follow-up care [16]. In our study, 26.79% of the respondents saw a GP once or twice prior to being referred to a hospital on suspicion of cancer, and 22.11% responded that they did not see a GP as they went straight to the hospital. After the first visit to the physician’s office about the health problem caused by their cancer, 61.42% of the respondents said it took less than a month to get diagnosed, and 25.69% said it took from one to three months. There were 2.11% of patients for which it took more than a year to get their diagnosis. Following the confirmation of a cancer diagnosis, 71.1% of the respondents reported that it took less than a month to start with their first oncology treatment, 25.16% said it took less than three months, and 3.74% said it took less than a year to start with the treatment. Overall, more than two-thirds (67.36%) of the respondents stated that they were satisfied with the length of time needed for getting tests done, 16.95% thought it took a little bit too long, and 13.5% responded that it took too long.

Patients’ experience in getting easy access to information.

Patient access to information is generally favoured, but it is still unclear whether providing patients access to their medical records improves healthcare quality [17]. Only one-third of the respondents (34.02%) claimed that they had their cancer treatment options explained in detail, 35.2% to some extent, and 19.19% were provided no explanation. Here is a comparison demonstrating how many times patients had consultations with their oncologist before they started with their oncology treatment based on their type of cancer: the highest number of consultations (more than three) was reported in patients with prostate cancer (33.33%), as well as oesophageal, stomach, pancreatic, liver, or gall bladder cancer (26.42%); two to three consultations were reported in patients with sarcoma (50%) and brain/CNS cancer (46.46%); and one consultation was reported in patients with urological (66.67%), head and neck (62.96), skin (61.11%), and lung cancer (47.22%) (Table 2).

A cancer treatment plan with information about a patient’s disease, the goal of treatment, the treatment options for the disease, possible side effects, and the expected length of treatment was provided for more than half (52.76%) of the patients, almost one-third (32.8%) did not get a cancer treatment plan, and 12.07% did not know what a cancer treatment plan is at all (Table 3). The following cross tabulation shows the relationship between treatment options (that were offered) and a lifestyle with cancer (advice and support in dealing with side effects of the treatment).

Patients’ involvement in discussions regarding the right treatment options and decisions about their care differed depending on the type of cancer. The largest frequency of negative response occurred in two-thirds of the respondents who reported that they were not involved in the decision regarding care and treatment when they desired to be involved, and the type of cancer of said patients included sarcoma (66.67%), oesophageal, stomach, pancreatic, liver, or gall bladder (64.15%), gynaecological (63.81%), and urological (60%) (Table 4.).

Patients’ experience in seeking and receiving care.

When asked whether they found someone from the hospital staff whom they could talk to about their worries and fears, less than half (45.93%) responded that they did not find someone from the hospital staff whom they could talk to, 35.20% responded ‘to some extent’, and only a small portion, i.e., 16.10% of the respondents, found someone whom they could talk to. As for the information on the support from health or social services (district nurses, home visits, psychological support), once again, over half of the respondents (54.63%) said that there was not enough care or support, and only a small portion of the respondents (6.26%) said they had been given adequate care, or received it at least to some extent (15.33%). Respondents were asked whether they were worried about their condition or treatment following the discharge, and most of them (57.48%) reported that they were not given any contact information, unlike 38.25% who stated that they were given the contact information in case of any concerns.

Patients’ experience in communication with healthcare providers.

When we asked participants to recall their experience of the delivery of bad news, more than two-thirds of the respondents (66.17%) claimed that they had not been told they could bring a family member or a friend with them, and only 19.59% were told they could do so. Bad news that was delivered by phone or discharge letter was experienced by 9.43% of the patients. Furthermore, over half of the respondents (52.97%) reported that the way they were told that they had cancer was sensitive, 22.6% reported that it should have been done ‘a lot’ more sensitively, and 24.43% respondents reported that it should have been done ‘a bit’ more sensitively. When asked to rate their overall cancer care, on a scale from 0 (very poor) to 10 (very good), only one-third (37.27%) of the respondents rated their care positively (7 or higher).

Patients’ preferences, needs, and values.

To be able to identify key points where the care and treatment may improve, the respondents were asked at which point they had felt the highest rate of inefficiency. They were offered 10 options and they could select all that applied to them. The largest share of respondents felt that the highest inefficiency was present in the communication between different HCPs involved in their cancer care (42.40%), dealing with the psychological impacts (48.21%), financial implications (33.25%), and ongoing side effects (32.76%). There was an option to select three out of four items when asked what needs to be the most improved in relation to their diagnosis. Most of the respondents felt that the greatest improvement would be to perform all services in one place (60.81%), to have a multidisciplinary team coordinated by one person (55.28%), and to have a shorter waiting time for the results (47.97%). It seems that the patients value the organisation and management of care, rather than the way their diagnosis is communicated to them (21.46%).

## 4. Discussion

The 69-item CPES adjusted for Central and Eastern European countries was distributed to 2460 purposively sampled cancer patients in Croatia in 2019. The study’s aim was to get input about cancer patients’ overall experience from the first suspicion of cancer, to oncology treatment, and to the end of life. In this paper, we did not report all of the obtained results from this survey, but highlighted four domains of a patient’s experience: getting timely appointments, easy access to information, seeking and receiving care, and communication with healthcare providers. We believe that, with the obtained data from the Croatian subgroup of cancer patients in this study, we have identified areas that are the most critical for improvement. Consensus needs to be reached between all healthcare professionals involved in providing care and support to oncology patients—oncologists, nurses, physiotherapists, psychologists, radiologists, primary care practitioners, social workers, and patient representatives. Finally, it would be suitable that this consensus results in interventions that can be applicable at a national level.

In the Croatian (CRO) subgroup sample, 45.53% responded that they were seen as soon as possible by a hospital doctor/oncologist, compared to 84% in ENG, 84% in NI, 81% in WLS, and 83% in SCT. For patients’ experience in getting diagnosed, in CRO, 26.79% responded that they saw a GP once or twice prior to being referred to a hospital on suspicion of cancer, and 22.11% of the respondents said that they went straight to the hospital (compared to 58% and 11% in ENG, 75% and 9.5% in NI, 61% and 4% in WLS, 77% and 23% in SCT, respectively).

Patients’ experience with the delivery of bad news reflects how they will adjust to their diagnosis and treatment. An inappropriate way of communicating with the patient can have a huge impact on the way they perceive their disease [18]. In Western countries, bad news was delivered sensitively to 86% of patients in ENG, 86% in NI, 84% in WLS, and 86% in SCT. However, in CRO, only 52.97% of the patients responded positively to this question. Related to the discussion and deciding on the best treatment, CRO patients had a less positive experience in this domain; only one-third (34.02%) responded that they received a complete explanation of treatment options before they started with the treatment, and only 31.46% were involved in decisions about their care and treatment (compared to 75% and 81% in ENG, 86% and 80% in NI, 97% and 78% in WLS, 87%, and 79% in SCT, respectively). Moreover, in CRO, the rate response was less positive, only 27.48%, regarding concerning possible side effects of treatment being explained in an understandable way, in comparison to 70% in ENG, 72% in NI, 75% in WLS, and 72% in SCT, respectively.

Cancer leads to major life changes for individuals: it affects the physical, psychological, social, and spiritual aspects of patients’ lives, as well as those of their families [19]. In all of these aspects it is necessary to provide cancer patients with support. In our study, one-third (38.25%) responded that they were informed of who to contact if they had any concerns about their condition or treatment after leaving the hospital, and only 6.26% definitely received enough care and support from health or social services during the treatment (for example, district nurses, home visits, psychological support) ( compared to 91% and 26% in ENG, 94% and 68% in NI, 89% and 59% in WLS, 95% and 60% in SCT, respectively).

The results of the present study indicate low levels of patient satisfaction with aspects of care during diagnosis and treatment planning. Croatian cancer patients’ experience in all domains is on a lower scale in comparison with England, Northern Ireland, Scotland, and Wales. Even though delivery of these results might not have a direct impact on disease management or outcomes in Croatian patients, it can affect the level of patient distress. The relevance of the patient–provider relationship, including effective communication, is recognised as a critical aspect of the patient experience [20]. Oncology patients need communication that allows them to feel guided, build trust, and sustain hope [21,22]. In the United States and the United Kingdom, the patient navigator role is a good demonstration of practice for guidance and support for cancer patients. It is designed to support patients in finding their way through health and social care systems, and to help them overcome barriers to accessing services. Pilot projects in Germany and France show that patient navigator programs are becoming more popular in other countries too, with some that focus on remote care provision introduced during the COVID-19 pandemic [23]. Moreover, another good example is the National Cancer Registration Service (NCRS), which has been collecting information on every patient diagnosed with a malignant tumour in England since 2012. The Patient Portal is a secure online system where patients can access the records on their condition held by the NCRS. Patients can access various types of information, including reports on tissue samples and scans, data on radio and chemotherapy, and information about their hospital attendance and treatments. Alongside this, the portal has a quality-of-life tool, a space for users to add their own notes, keep a list of contacts, and access a directory of links to helpful information on support, treatment, clinical trials, and research [24].

Achieving patient-centred care requires a multidisciplinary approach that facilitates effective communication by ensuring that all team members are familiar with the patient’s history and are involved in conceptualizing the treatment plan [25]. Based on the results of this study, communication skills workshops were launched for all oncology health professionals in Croatia, with a focus on the issues found in this research. Also, with the aim of better informing medical doctors, other health professionals, and oncology patients, it is necessary to continue to develop platforms with available information, exchange of experiences, and organisation of targeted education. Clinical nurse specialists/oncology nurses are a crucial part of a multidisciplinary team [26,27]. Oncology nurses ensure that the wishes and needs of patients are respected during the process of diagnosis, therapy, and follow-up [28]. Many European countries, including Croatia, still lack formal oncology education programs that could prepare nurses to provide a higher level of oncology care. The implementation of the educational framework for oncology nurses [29] in Croatia is of utmost importance for the minimum set of standards for oncology-related knowledge and skills required for the care of cancer patients.

The subjects of this study were members of a coalition of oncology patients’ associations because, in this way, we ensured that the data were filled in by proven oncology patients. This may be a limitation of the study, as well as the fact that there was a greater representation of women’s associations (associations of breast cancer and gynaecological cancer patients) in the coalition of oncology patients’ associations. According to available data, the incidence of cancer in Croatia in 2019 amounted to 13,547 new cases of cancer in men and 11,805 new cases of cancer in women [30].

## 5. Conclusions

This study adds to the current understanding of cancer patient experience in Croatia and, based on the obtained data, there is significant room for improvement. The obtained data from this study provided insight into the areas of oncology care in Croatia that are in need of improvement. With this study, we emphasised the need for specific knowledge of healthcare professionals, but we were also committed to continuing to evaluate the experiences of cancer patients.

## Figures and Tables

**Table 1 ijerph-19-08285-t001:** Frequency distribution according to the type of cancer.

Type of Cancer	Frequency	Percentage
Brain/central nervous system	31	1.3
Skin	108	4.4
Oesophageal, stomach, pancreatic, liver, or gall bladder	83	3.4
Urological	52	2.1
Breast	960	39
Colorectal/bowel	218	8.9
Gynaecological	223	9.1
Haematological	163	6.6
Head and neck	98	4
Lung	472	19.2
Prostate	43	1.7
Sarcoma	9	0.4
Total	2460	100

**Table 2 ijerph-19-08285-t002:** Cross tabulation of patients’ consultations with their oncologist before they started with their oncology treatment based on their type of cancer.

		Before Your Cancer Treatment Started, How Many Consultations Did You Have with the Physician Who Runs Your Treatment?									
		One		Two to Three		More than Three		None		Total	
		N	%	N	%	N	%	N	%	N	%
Type of cancer	Brain/central nervous system	4	36.36	5	46.46	0	0	2	18.18	11	0.89
	Breast	268	58.52	68	14.85	44	9.61	78	17.03	458	37.21
	Colorectal/bowel	42	34.15	48	39.02	6	4.88	27	21.95	123	9.99
	Gynaecological	32	30.48	43	40.95	4	3.81	26	24.76	105	8.53
	Haematological	26	32.91	23	29.11	10	12.66	20	25.32	79	6.42
	Head and neck	34	62.96	12	22.22	2	3.7	6	11.11	54	4.39
	Lung	119	47.22	56	22.22	34	13.49	43	17.06	252	20.47
	Prostate	7	33.33	7	33.33	7	33.33	0	0	21	1.71
	Sarcoma	2	33.33	3	50	1	16.67	0	0	6	0.49
	Skin	33	61.11	12	22.22	0	0	9	16.67	54	4.39
	Oesophageal, stomach, pancreatic, liver, or gall bladder	25	41.17	7	13.21	14	26.42	7	13.21	53	4.31
	Urological	10	66.67	5	33.33	0	0	0	0	15	1.22
Total		602	48.9	289	23.48	122	9.91	218	17.71	1231	100

**Table 3 ijerph-19-08285-t003:** Cross tabulation of the offered treatment options, advice, and support in dealing with side effects of treatment(s).

		Before Your Cancer Treatment Started, Were Your Treatment Options Explained to You?											
		Yes, Completely		Yes, to Some Extent		No		There Was Only One Type of Treatment That Was Suitable for Me		Don’t Know/Can’t Remember		Total	
		N	%	N	%	N	%	N	%	N	%	N	%
Were you offered practical advice and support in dealing with the side effects of your treatment(s)?	Yes, definitely	358	14.55	55	2.24	6	0.24	16	0.65	0	0	435	17.68
	Yes, to some extent	338	13.74	422	17.15	123	5	143	5.81	6	0.24	1032	41.95
	No, I was not offered any practical advice or support	113	4.59	389	15.81	329	13.37	109	4.43	11	0.45	951	38.66
	Don’t know/can’t remember	28	1.14	0	0	14	0.57	0	0	0	0	42	1.71
Total		837	34.02	866	35.2	472	19.18	268	10.89	17	0.69	2462	100

**Table 4 ijerph-19-08285-t004:** Cross tabulation of the patients’ desired involvement in decisions about their care and treatment depending on their type of cancer.

		Were You Involved as Much as You Wanted to Be in Decisions about Your Care and Treatment?									
		Yes, Definitely		Yes, to Some Extent		No, but I Would Like to Have Been More Involved		Don’t Know/Can’t Remember		Total	
		N	%	N	%	N	%	N	%	N	%
Type of cancer	Brain/central nervous system	0	0	9	81.82	2	18.18	0	0	11	0.89
	Breast	56	12.23	161	35.16	223	48.69	18	3.93	458	37.21
	Colorectal/bowel	12	9.76	51	41.46	60	48.78	0	0	123	9.99
	Gynaecological	17	16.19	14	13.33	67	63.81	7	6.67	105	8.53
	Haematological	10	12.66	49	62.03	17	21.52	3	3.80	79	6.42
	Head and neck	17	31.48	27	50	10	18.52	0	0.0	54	4.39
	Lung	47	18.65	91	36.11	111	44.05	3	1.19	252	20.47
	Prostate	0	0	13	61.9	8	38.1	0	0	21	1.71
	Sarcoma	2	33.33	0	0.0	4	66.67	0	0	6	0.49
	Skin	0	0	25	46.3	29	53.7	0	0	54	4.39
	Oesophageal, stomach, pancreatic, liver, or gall bladder	7	13.21	6	11.32	34	64.15	6	11.32	53	4.31
	Urological	6	40	0	0.0	9	60	0	0	15	1.22
Total		174	14.54	446	37.26	574	47.95	37	0.25	1231	100

## Data Availability

Data supporting reported results can be found at Croatian Coalition of Health Association.

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
