# Peer review of "Croatian National Cancer Patient Experience Survey"

_ijerph, 2022, doi:10.3390/ijerph19148285_

Round 1

Reviewer 1 Report

This is an interesting paper and I think patient experience is an important and relevant topic.  I thought it was well written.  Minor suggestions to consider:

1.  Spacing I thought spaces should be added e.g. line 145, 162, 175, 183, 194, 205, 237, 257 & 265.

2.  Line 56-86 is quite long & I think it should be broken up into 2 paragraphs.

3.  Line 177 - I would add the word 'reported' after respondents.

4.  Page 4 last paragraph - I notice lung cancer is not mentioned here but is in the table below.  I would add it.

5.  Page 4  line 192- I would add the word 'reported' after 57.68%.

6.  Reference is needed line 357.

Author Response

Dear Reviewer,

Thank you for the valuable comments. We applied all of the reviewer suggestions and explained all of the changes in this document. Throughout the manuscript we used a red-coloured font in order to make it easier to track changes; we hope that this will be acceptable.

Reviewer 2 Report

I found this a very interesting paper - however the English grammar editing was too extensive to do a comment line by line - many missing articles and general grammer problems.  I would be glad to edit a word version if provided as I think it should be published.  

Author Response

(The authors gave the same response as above.)

Reviewer 3 Report

This is an interesting article looking at the experience and perspectives of patients receiving cancer care in Croatia using a questionnaire that has been used across several countries. There are some areas that are suggested to strengthen the paper: 

Abstract line 27-28: it is suggested the 'most of the respondents felt ...' when the number reporting this was 61% and 55% respectively and this reviewer does not consider this to be "most" and suggest that the authors change this 

Literature review line 62 to 82: the information about the results from the various UK surveys are not needed in the literature review and are repeated in the discussion section - from line 233 onwards. It is suggested that the authors include this information in the discussion and remove from the literature review 

Table 1 and discussion from line 103 to 110 appear to be results and therefore should be moved into the results section.

This reviewer is concerned that there did not appear to be any ethics approval sought from a relevant committee. Can the authors please confirm this 

Line 125 and 126 are saying exactly the same thing. This reviewer recommends that this is rewritten into one sentence 

This reviewer found the results section confusing as it is a repeat of the information that is in the tables. The numbers in Table 2 and Table 4 based on the type of cancer get very small and this reviewer is not sure that this adds to the story. Perhaps using the number of consultations and involvement in decisions overall rather than breaking this down further would be more meaningful? 

It is not clear why the data was analysed as a cross-tabulation (in Table 3). Is there a relationship between treatment options explained and being offered practical advice? This is not clear to this reviewer and it is recommended that this be explained in the paper. 

The discussion appears to be repeating much of the results. This reviewer would have appreciated some discussion around some of the things that are in place in the UK that may be used in Croatia as there appears to be a higher level of positive experience among patients. In addition, are there other countries in Europe that may be used to compare these results to? Is there something to do with the health care system in the UK as compared to other countries in Europe? Are there any other recommendations apart from an "educational framework for nurses"? Is there anything that can assist medical doctors being more informed about these results to improve the overall treatment of cancer patients? 

There do not appear to be any limitations to the study included in the paper. Some that may be important for discussion include only those who were part of a membership were invited to participate - what biases would this have introduced? There appears to be an overrepresentation of females in the sample; some information related to the representativeness of the sample to cancer patients in Croatia would have been helpful; and what was the response rate for the survey? 

Author Response

(The authors gave the same response as above.)

Round 2

Reviewer 3 Report

I would like to acknowledge that the authors have taken on board the suggestions from all the reviewers and this has greatly strengthened the paper. This reviewer feels that the paper is now ready for publication.